

# How to tune the absorption spectrum of chlorophylls to enable better use of the available solar spectrum

Pedro J. Silva[1,2], Maria Osswald-Claro[3] and Rosário Castro Mendonça[3]

[1] UCIBIO@REQUIMTE, BioSIM, Departamento de Biomedicina, Faculdade de Medicina, Universidade do Porto, Porto, Portugal
[2] FP-I3ID,FP-BHS, Fac. de Ciências da Saúde, Universidade Fernando Pessoa, Porto, Portugal
[3] Deutsche Schule zu Porto, Porto, Portugal

Corresponding author
Pedro J. Silva, pedros@ufp.edu.pt

## ABSTRACT

Photon capture by chlorophylls and other chromophores in light-harvesting complexes and photosystems is the driving force behind the light reactions of photosynthesis. Excitation of photosystem II allows it to receive electrons from the water-oxidizing oxygen-evolution complex and to transfer them to an electron-transport chain that generates a transmembrane electrochemical gradient and ultimately reduces plastocyanin, which donates its electron to photosystem I. Subsequently, excitation of photosystem I leads to electron transfer to a ferredoxin which can either reduce plastocyanin again (in so-called ''cyclical electron-flow'') and release energy for the maintenance of the electrochemical gradient, or reduce $NADP^+$ to NADPH. Although photons in the far-red (700–750 nm) portion of the solar spectrum carry enough energy to enable the functioning of the photosynthetic electron-transfer chain, most extant photosystems cannot usually take advantage of them due to only absorbing light with shorter wavelengths. In this work, we used computational methods to characterize the spectral and redox properties of 49 chlorophyll derivatives, with the aim of finding suitable candidates for incorporation into synthetic organisms with increased ability to use far-red photons. The data offer a simple and elegant explanation for the evolutionary selection of chlorophylls *a*, *b*, *c*, and *d* among all easily-synthesized singly-substituted chlorophylls, and identified one novel candidate (2,12-diformyl chlorophyll *a*) with an absorption peak shifted 79 nm into the far-red (relative to chlorophyll *a*) with redox characteristics fully suitable to its possible incorporation into photosystem I (though not photosystem II). chlorophyll *d* is shown by our data to be the most suitable candidate for incorporation into far-red utilizing photosystem II, and several candidates were found with red-shifted Soret bands that allow the capture of larger amounts of blue and green light by light harvesting complexes.

## INTRODUCTION

Chlorophylls are produced from protoporphyrin IX (*Willows, 2003*) through the sequential insertion of $Mg^{2+}$, esterification of the propionyl group in ring C with a methyl group, oxidative cyclization of this substituent to generate a new ring E, and reduction of

a vinyl group (Fig. 1). Depending on the organism, the intermediate thus formed (protochlorophyllide $a$) may undergo a reduction of the double bond between carbons $C_{17}$ and $C_{18}$, yielding chlorophyllide $a$ (from which chlorophyllides $b$, $d$, and $f$ may be produced through the oxidation of the methyl groups on $C_7$, $C_3$ or $C_2$, respectively) or alternatively the propionyl group on $C_{17}$ may be oxidized to a propenyl group (yielding chlorophylls $c_1$ or $c_2$, depending on the absence or presence of subsequent reactions which modify the ethyl sidechain on ring B). Finally, addition of phytyl chains to the $C_{17}$ propionyl groups of the chlorophyllides converts them into the corresponding chlorophylls. The large variety of possible modifications to the porphyrin (or chlorin) ring leads to diverse absorption spectra and fluorescence behaviour, as reviewed by *Taniguchi & Lindsey (2021)*. The determination of the intrinsic color of chlorophylls is not, however, a straightforward endeavor due to their strong interactions with solvent molecules. I*n vivo,* water molecules coordinate their magnesium central ions and in purified chlorophyll extracts a similar role is fulfilled by polar solvents such as acetone and ethyl acetate, which also interact with the polarizable $\pi$-cloud in the heterocyclic ring. As a result, the intense absorption bands around 430 nm in chlorophylls $a$ and $b$ (the B-bands, or Soret bands) can shift their maxima by up to 15–20 nm and the Q-bands (around 650 nm) can shift up to 10 nm when solvent is varied from acetone to pyridine (*Taniguchi & Lindsey, 2021*). In the absence of solvent, the absorption bands are blue-shifted to wavelengths below 420 nm (Soret band), or between 626–636 nm (Q-band) (*Milne et al., 2015*). The Q-band absorptions in the 600-700 nm are crucial for the generation of the very-low potential states of photosytems I and II responsible for feeding electrons into the photosynthetic electron-transport chain, whereas the absorptions in the Soret-band play an important role in the capture of light in the antenna complexes. Early experimental and theoretical work has established that two distinct Soret absorptions and two Q-bands are actually present (*Gouterman, 1978*; *Weiss, 1978*) (corresponding to two different polarizations) and the balance between their relative intensities and energies can give rise to a single symmetrical absorption peak in each region, to an asymmetrical peak featuring a red (or blue-)-shifted shoulder, or to two separate peaks at each region.

The solar spectrum that reaches the surface of the Earth differs from the theoretical blackbody spectrum due to the selective absorption of specific wavelengths by atoms in the solar atmosphere, to scattering phenomena in the atmosphere and to absorption by components of the Earth's atmosphere. The largest deviations from the blackbody spectrum lie in the absorption of most of the ultra-violet radiation by atmospheric ozone, and in the selective depletion of portions of the infra-red spectrum due to absorption by water and $CO_2$. In the visible portion of the spectrum, the number of photons of a given wavelength increases about three-fold as the wavelength is increased from 360 nm to 440 nm, then increases about 30% as the wavelength is increased to 520 nm and then (apart from pontual sharp decreases due to depletion of some specific wavelengths) remains roughly constant at all wavelengths until 800 nm (Fig. 2). Analysis of the number of photons with energy above that of a specific wavelength shows an approximately linear relationship for wavelengths above 460 nm (Fig. 2, dotted trace). As a consequence, increasing the wavelength at which a molecule absorbs light in this spectral region may allow (if a mechanism to

**Figure 1   Biosynthetic pathway of chlorophyllide biosynthesis from protoporphyrin IX.**

funnel photons of lower wavelengths to this absorption is present) a notable increase in photochemical efficiency. One can therefore envisage that specific chemical modifications of the chlorophyl substituents may change their spectral properties and allow them, for example, to capture additional photons with lower energies in the antenna complexes (by shifting the Soret-bands to higher wavelengths) or to enable better capture of 680-700 nm photons by photosystems I and II, as observed experimentally for the chlorophyll *d*-containing far-red-light-adapted photosystem I from *Acaryochloris marina* (*Kimura et al., 2022*).

In this work, we have used computational methods to obtain the gas-phase UV-Vis spectra of a large number of chlorophyll derivatives, in order to gather insight on the influence of specific substituents on the spectral properties of this class of compounds. We analyzed the derivatives obtained through oxidation of $CH_3$ groups to formyl (as catalyzed by chlorophyllide *a* oxidase (*Eggink et al., 2004*) or related enzymes with different subtrate binding orientation), as well as those obtained replacement of methyl/ethyl groups by vinyl or replacement of vinyl groups by methyl. Although the latter modifications are more complex than the oxidation of methyl to formyl, they are not *a priori* prevented by the chemical characteristics of the reactions catalyzed by the corresponding enzymes, since they can be accessed through engineered biosynthetic pathways involving mutant forms of uroporphyrinogen decarboxylase and coproporphyrinogen oxidase with altered active sites that accommodate substrates with a different arrangement of acetate/methyl/propionyl

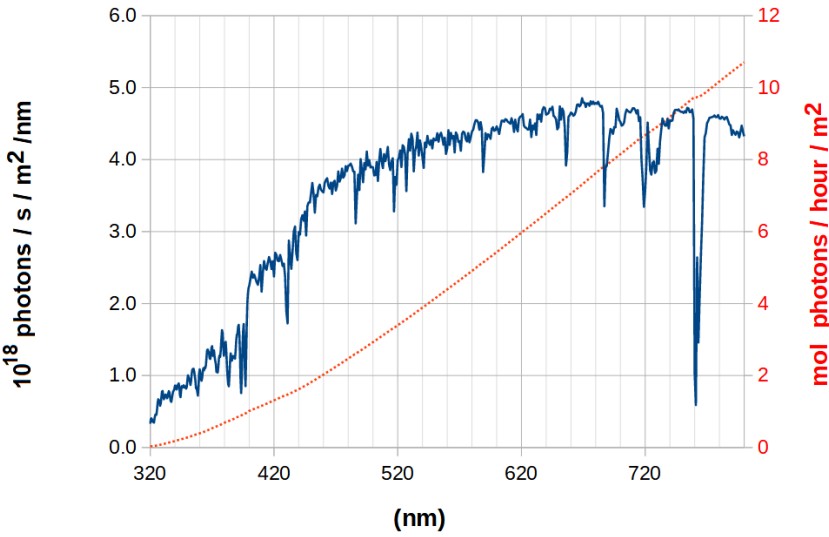

**Figure 2** **Surface solar spectrum (blue, continuous line) and its integration, the number of photons with energy above that of a given wavelength (red, dotted line, right axis).** Data computed from the reference spectrum ASTM G-173-03 (*International Standard Organization, 1992*).

substituents around the tetrapyrrole ring. Our computations reveal the influence of chlorophyll modifications on their intrinsic spectra, identify which derivatives are most suited for incorporation into higher-efficiency photosystems or light-harvesting complexes, and suggest plausible biosynthetic routes to engineer their incorporation into modified organisms.

## Computational methods

All quantum chemistry computations were performed with the Firefly (*Granovsky, 2013*) quantum chemistry package, which is partially based on the GAMESS (US) (*Schmidt et al., 1993*) source code. DFT and TDDFT computations were performed using the 6-31G* basis set with the B3LYP functional (*Lee, Yang & Parr, 1988*; *Becke, 1993*; *Hertwig & Koch, 1995*), which has been shown previously to provide good results when applied to the analysis of excited states of chlorophyl *a* (*Sirohiwal et al., 2020*). For efficient computation, the modifications of the $C_{17}$-propionyl sidechain by phytyl observed in the conversion of chlorophyllides to chlorophylls were not included in our model because this portion of the molecules is electronically segregated from the delocalized $\pi$-system and is, therefore, irrelevant for the electronic transitions leading to the UV-Vis absorption of chlorophylls. Autogenerated delocalized coordinates (*Baker, Kessi & Delley, 1996*) were used for geometry optimizations in the ground state. Up to nine excited states were then computed at the optimized geometry using TDDFT. Spectra were simulated in Chemcraft (Gaussian broadening, 30 nm width at half-height) using the computed TDDFT excitation energies and oscillator strengths. Redox potentials were computed from the difference in energies between the geometry-optimized neutral molecules and their (geometry-optimized) one-electron-oxidized forms at the B3LYP/6-31G(d) level using the polarizable continuum method implemented in Firefly, with a dielectric constant of 20.

## RESULTS

Complete agreement between computed vertical excitation energies and absorption maxima cannot be expected to be obtained for several reasons. First, experimental absorption maxima are not exactly the same as vertical excitation energies computed in the gas phase because the solvent used in the experimental determination affects the ground and the excited states to different extents. Moreover, vibrational effects entail that absorption bands have a finite width (rather than occurring at a single wavelength) and if several transitions occur in close proximity it is possible that the maximum absorption is observed at a wavelength that does not correspond to that of any individual transition, but to a point where both bands are absorbing with large contributions. When individual bands are sufficiently separated this source of error becomes quite small, but the overlap of the experimental Soret bands in most chlorophylls renders the attribution of an absorption band to a specific predicted transition less secure. We therefore evaluated the adequacy of the chosen computational method through the comparison of the spectrum computed from the calculated TD-DFT excitation energies and oscillator strengths with the experimental gas phase spectral data (Table 1). Although the absorption bands are predicted by TDDFT to lie at lower wavelengths (*i.e.,* higher energies), the experimental trends are very well conserved, which shows that the chosen theory level can reliably predict whether any given modification will shift absorption to higher or lower energies, as well as a good estimate of the quantitative shift in excitation energy (Fig. 3). The spectra predicted in the presence of an acetone polarizable continuum method contain more structure than the ones computed in the gas-phase, but the fit of absorption maxima has worse $R^2$-values than that obtained with the gas-phase results. Since the inclusion of solvent in our computations not only did not afford a sufficient improvement above the gas-phase computations, but actually decreased the goodness of the fit, all the remaining TDDFT-computations were performed using only gas-phase methodology.

Linear regression of actual *vs.* TDDFT-computed gas phase absorption peaks returns the following relationships, which can be used to obtain corrected absorption maxima from the TDDFT-predicted absorption peaks:

Soret-band: actual peak (eV) $= 0.587 \times$ TDDFT-computed absorption peak (eV) $+ 0.961$

Q-band: actual peak (eV) $= 1.131 \times$ TDDFT-computed absorption peak (eV) $- 0.544$

This comparison with experimental spectra also shows that TDDFT systematically under-predicts the relative intensity of the Q-bands. This is, however, mitigated by the good correlation between predicted and actual absorption strength: in every comparison between the relative strength of Q-bands between two chlorophylls, TDDFT-predicted increases/decreases in Q-band intensity systematically correspond to the experimental observations. Comparison of predicted and experimental Soret bands for chlorophylls *d* and *f* reveal one shortcoming of the chosen method: the energetic difference between the two absorptions in the Soret-band is predicted to amount to no more than $0.10-0.16$ eV, (rather than the experimentally observed $0.30-0.38$ eV), thus conflating the two separate Soret bands in these chlorophylls into a single one. Despite this, TDDFT correctly predicts that in chlorophylls *d* and *f* the two Soret-bands have similar oscillator strengths, in

**Table 1  Comparison of the main features of the experimental acetone spectra with the spectra simulated (Gaussian broadening, 30 nm width at half-height) from the gas-phase TDDFT-computed excitation energies and oscillator strengths.** Relative intensities are shown in parentheses.

| Molecule | Experimental Soret bands in acetone (nm) | Computed Soret-bands (nm) | Experimental Q-bands in acetone (nm) | Computed Q-bands (nm) |
|---|---|---|---|---|
| Chlorophyll *a* | 430.3 (100) | 378.5 (100) | 662.1 (82) | 578.8 (25) |
| Chlorophyll *b* | 456.9 (100) | 415.4 (100) | 645.5 (35) | 569.7 (15) |
| Chlorophyll *d* | 393 (72) 447 (100) | 399.3 (100) | 688 (116) | 600.6 (34) 559.6 (6) |
| Chlorophyll *f* | 397 (100) 439.5 (81) | 392.6 (100) | 698 (129) | 601.8 (42) 559.5 (7.8) |

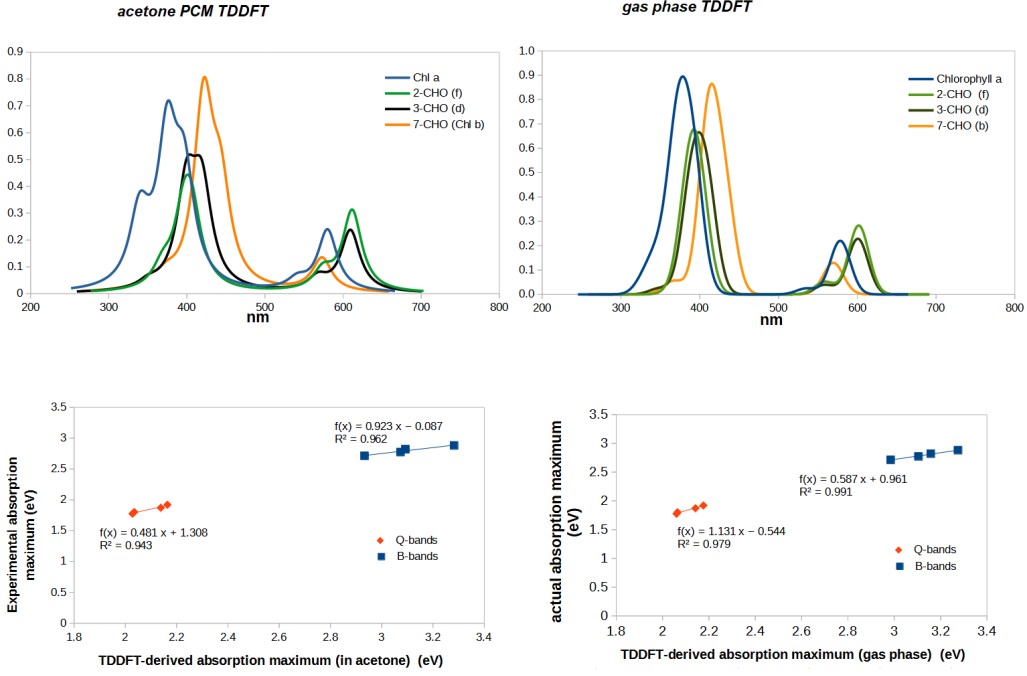

**Figure 3  Correlation between experimental and predicted absorption maxima of the B- and Q-bands on chlorophylls *a*, *b*, *d*, and *f*.** For molecules with two separated absorption bands in the same region, the predicted absorption band was plotted against the wavelength of the lower-energy experimental absorption.

contrast to chlorophylls *a* and *b*, where one of them is 1.5 −2× more intense. In summary, TDDFT with the B3LYP functional and the 6-31G(d) affords a very good performance in the (low-energy) Q-band absorptions (which are the ones more relevant to the chlorophyl role in photosystems I and II) and a less perfect behavior in the prediction of the features of the (higher energy) Soret-bands.

Having established the appropriateness of the chosen theoretical level, we now analyze the predicted spectra of the fifteen single-substitututed chlorophylls studied, in each case comparing them to that of chlorophyll *a* (Fig. 4 and Table 2). It is evident from the data

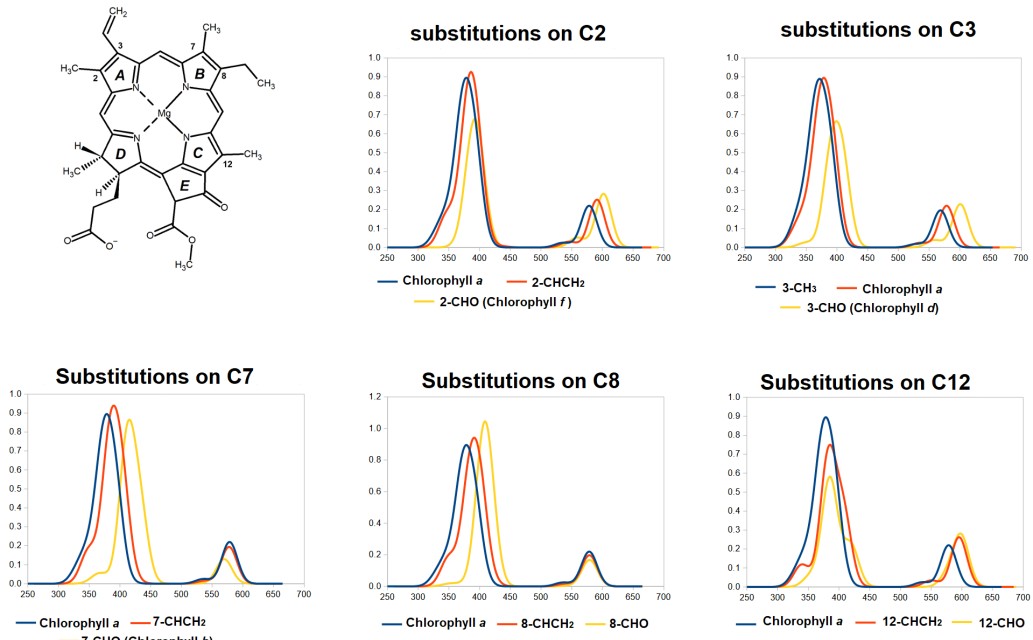

**Figure 4** **TDDFT-predicted spectra (PBE0/6-31G*) of selected derivatives of chlorophyll *a*.** In each graph, the spectrum of the derivative bearing a methyl (or ethyl, for position 8) at the given position is depicted in blue. Spectra of derivatives bearing a vinyl group at that position are depicted in red, and those of molecules bearing formyl at that position are depicted in yellow.

in Fig. 4 that the spectral consequences of replacement of methyl by vinyl (-CH $=$ CH$_2$) or formyl (CHO) are remarkably dependent on the precise positioning of the substitution. Introduction of a vinyl group at positions 2- or 3- shifts both the Soret band and the Q-band to lower energies (higher wavelengths). At these positions, formyl causes larger shifts of both absorptions to lower energies. At positions 7- and 8-, both vinylation and formylation blue-shift the Soret band but have either no effect at all on the Q-band (for vinylation at position 7 or formylation at either position) or slightly shift it to a lower wavelength. At position 12-, vinilyation and formylation cause equal blue-shifts to the Q-band absorption and cause modest blue-shift on the Soret-band, akin to those observed with equivalent substitutions at the 2- position but with a larger separation of the two Soret band absorptions (0.21 –0.26 eV). In all studied positions, vinylation increases the absorption wavelenght of the Soret-band by moderate amounts (6–13 nm) but only causes significant shifts (13–17 nm) at the Q-band absorption when it occurs at the 2- or 12- position. In contrast, and depending on the modification site, formylation has widely disparate effects on both the Soret and Q-bands and changes the separation between Soret and Q- absorption maxima from 200 nm (chlorophyll *a*) to as little as 154 nm (in chlorophyll *b*, which bears a formyl on carbon 7) or as much as 223 nm (in the 12-formylated derivative).

Interestingly, the two single modification to chlorophyll *a* that yield the largest predicted red-shift in Q-band absorption are the ones that yield the naturally-occurring pigments

**Table 2  TDDFT-computed absorption maxima of the Soret and Q-bands upon single substitutions of chlorophyll _a_.** The largest deviations in band position are highlighted in bold.

| Formylation at position... | Soret band maximum (nm) | Q-band maximum (nm) | Vinylation at position... | Soret band maximum (nm) | Q-band maximum (nm) |
|---|---|---|---|---|---|
| 2- (Chl _f_ ) | 392.7 | **601.9** | 2- | 386.1 | 591.3 |
| 3- (Chl _d_ ) | 398.9 | **600.7** | 3- (Chl _a_) | 378.5 | 578.5 |
| 7- (Chl _b_ ) | **415.4** | 569.7 | 7- | 390 | 577.4 |
| 8- | 408.8 | 579.1 | 8- | 391.3 | 578.8 |
| 12- | 373.9 | 597.5 | 12- | 385.4 | 595.3 |

chlorophyll _f_ and _d_, which are produced in organisms that live in conditions where light is enriched in large-wavelength/far-red photons (_Chen, 2019_; _Chen & Blankenship, 2011_). This observation shows that evolution has been able efficiently search the chemical space of such single-site modifications to select the most appropriate derivatives to optimize the absorption of the low-energy visible photons. Conversely, the single-substituted variant of chlorophyll _a_ that yields the largest red-shift in the Soret band (responsible for the acquisition of photons in the antenna) is the formylation in position 7, which yields the abundant photosynthetic pigment chlorophyll _b_, present in oxygenic photosynthetic organisms and which is required for the correct assembly of light-harvesting complexes (_Tanaka & Tanaka, 2011_; _Voitsekhovskaja & Tyutereva, 2015_). It thus appears that evolution has also been able to explore the accessible single-modification space of chlorophyll _a_ to find the most appropriate single-substituted derivative for use in light-harvesting complexes.

Since it is likely that the introduction of additional modifications into the chlorophyll core will enable the capture of other portions of the light spectrum, we also studied the influence of double modifications of chlorophyll _a_ on its spectral features. As before, we analyzed the influence of vinylation, which can be accomplished through the activity of coproporphyrinogen oxidase on appropriately-located ethyl substituents, and formylation, which can be performed either through the action of chlorophyllide oxidase on methyl substituents (_Eggink et al., 2004_) or through the oxydation of vinyl substituents by molecular $O_2$ catalyzed by as-yet uncharacterized enzymes (_Schliep et al., 2010_). The results obtained are rather diverse (Tables 3 and 4), but several general trends can nonetheless be perceived and will be discussed below.

The Soret and Q-bands in the computed spectra for the double-substituted chlorophylls were almost invariably red-shifted. Exceptions to this trend were observed for the Q-bands of 7-formyl-chlorophyll bearing a second substituent on position 8 and in the two derivatives where the 3-vinyl group in chlorophyll _a_ was replaced by a methyl group. In these four cases, the Q-band was predicted to be moderately blue-shifted by only 5-9 nm. The Soret bands in all 34 double-substituted derivatives of chlorophyll _a_ studied were always red-shifted (by 7–63 nm). In molecules containing two additional vinyl groups this red-shifted was small (12–18 nm), whereas in molecules bearing two additional formyl groups the Soret band shift was the largest (21–63 nm). On average, the smallest effects

**Table 3  Predicted wavelengths of the maxima of Soret and Q-bands of double-substituted chlorophylls bearing at least one additional vinyl group.** Band positions were computed by taking the positions of the maxima of the TDDFT-computed spectra (B3LYP/6-311G(d,p) and transforming them according to the linear relationships between experimental and computational maximum absorption energies described in the first paragraph of the Results section. (sh.: shoulder.)

| | Soret band (nm) | Q-band (nm) | Redox potential (V) vs. $Chl_a^+/Chl_a$ | | Soret band (nm) | Q-band (nm) | Redox potential (V) vs. $Chl_a^+/Chl_a$ |
|---|---|---|---|---|---|---|---|
| **2-vinyl-** | | | | **2-vinyl-** | | | |
| 3-vinyl | 436 | 679 | 0.03 | 3-formyl | 447 | 714 | 0.12 |
| 7-vinyl | 445 | 677 | 0.07 | 7-formyl | 463 | 663 | 0.21 |
| 8-vinyl | 447 | 679 | 0.06 | 8-formyl | 439 | 674 | 0.19 |
| 12-vinyl | 443 | 711 | 0.04 | 12-formyl | 442 | 706 | 0.30 |
| **7-vinyl-** | | | | **7-vinyl-** | | | |
| 2-vinyl | 445 | 677 | 0.07 | 2-formyl | 449 | 693 | 0.22 |
| 3-vinyl | 439 | 658 | 0.04 | 3-formyl | 454 | 696 | 0.13 |
| 8-vinyl | 448 | 662 | 0.07 | 8-formyl | 461 | 669 | 0.20 |
| 12-vinyl | 443 | 691 | 0.05 | 12-formyl | 442 (sh. 464) | 690 | 0.31 |
| **8-vinyl-** | | | | **8-vinyl-** | | | |
| 2-vinyl | 447 | 679 | 0.06 | 2-formyl | 450 | 695 | 0.22 |
| 3-methyl | 437 | 648 | 0.00 | 3-formyl | 453 | 692 | 0.13 |
| 3-vinyl | 440 | 660 | 0.04 | 7-formyl | 464 | 655 | 0.21 |
| 7-vinyl | 448 | 662 | 0.07 | 12-formyl | 441 | 695 | 0.31 |
| 12-vinyl | 443 | 688 | 0.07 | | | | |
| **12-vinyl** | | | | **12-vinyl** | | | |
| 2-vinyl | 443 | 711 | 0.04 | 2-formyl | 449 | 721 | 0.34 |
| 3-vinyl | 435 | 685 | 0.04 | 3-formyl | 454 | 717 | 0.11 |
| 7-vinyl | 443 | 691 | 0.05 | 7-formyl | 465 | 668 | 0.22 |
| 8-vinyl | 443 | 688 | 0.07 | 8-formyl | 460 | 685 | 0.19 |

were seen in the addition of an extra vinyl group to 2-vinyl- or 12-vinyl-chlorophyll *a* and the largest effects were seen with the addition of a second formyl group to 7-formyl- or 8-formyl-chlorophyll *a*, which caused red-shifts above 35 nm. Accordingy, 7,8-diformyl-chlorophyll *a* presented the longest-wavelength predicted Soret bands, centered at 469 nm and 491 nm, which places it as the most attractive candidate to expand the range of high-energy photons captured in the light-harvesting complexes.

In the doubly-substituted chlorophylls, the Q-band maxima was observed to vary in an 84 nm-wide window. Usually, modifiying a singly-substituted chlorophyll through the addition of a formyl or vinyl group at the $C_{12}$-position resulted in larger red-shifts of the Q-band than similar additions at any of the other positions, and modifications on positions 7- and 8- had hardly any effect on the Q-band position relative to that of the corresponding singly-substituted chlorophylls. Since the modifications of the $C_7$- and $C_8$- usually caused the largest red-shifts in the Soret band positions, this confluence of factors leads the observed distance between the positions of the Soret and Q.bands in these double $C_7$-/$C_8$-derivatives to become much shorter than the 229 nm observed in chlorophyll *a*, and to become as close as as 182 nm (in 7-formyl-8-formyl-chlorophyll).

**Table 4  Predicted wavelengths of the maxima of Soret and Q-bands of double-substituted chlorophylls bearing at least one additional formyl group.** Band positions were computed by taking the positions of the maxima of the TDDFT-computed spectra (B3LYP/6-311G(d,p) and transforming them according to the linear relationships between experimental and computational maximum absorption energies described in the first paragraph of the Results section. (sh.:shoulder).

| | Soret band (nm) | Q-band (nm) | Redox potential (V) vs. $Chl_a^+/Chl_a$ | | Soret band (nm) | Q-band (nm) | Redox potential (V) vs. $Chl_a^+/Chl_a$ |
|---|---|---|---|---|---|---|---|
| **2-formyl-** | | | | **2-formyl-** | | | |
| 3-vinyl | 441 | 694 | 0.18 | 3-formyl | 451 | 713 | 0.34 |
| 7-vinyl | 449 | 693 | 0.22 | 7-formyl | 468 | 684 | 0.35 |
| 8-vinyl | 450 | 695 | 0.22 | 8-formyl | 467 | 690 | 0.33 |
| 12-vinyl | 449 | 721 | 0.34 | 12-formyl | 454 | 734 | 0.40 |
| **3-formyl-** | | | | **3-formyl-** | | | |
| 2-vinyl | 447 | 714 | 0.12 | 2-formyl | 451 | 713 | 0.34 |
| 7-vinyl | 454 | 696 | 0.13 | 7-formyl | 464 | 676 | 0.27 |
| 8-vinyl | 453 | 692 | 0.13 | 8-formyl | 465 | 686 | 0.26 |
| 12-vinyl | 454 | 717 | 0.11 | 12-formyl | 455 | 716 | 0.36 |
| **7-formyl-** | | | | **7-formyl-** | | | |
| 2-vinyl | 463 | 663 | 0.21 | 2-formyl | 468 | 684 | 0.35 |
| 3-vinyl | 457 | 647 | 0.20 | 3-formyl | 464 | 676 | 0.27 |
| 8-vinyl | 464 | 655 | 0.21 | 8-formyl | 469 / 491 | 651 | 0.41 |
| 12-vinyl | 465 | 668 | 0.22 | 12-formyl | 464 | 685 | 0.42 |
| **8-formyl-** | | | | **8-formyl-** | | | |
| 2-vinyl | 439 | 674 | 0.19 | 2-formyl | 467 | 690 | 0.33 |
| 3-CH3 | 448 | 650 | 0.11 | 3-formyl | 465 | 686 | 0.26 |
| 7-vinyl | 461 | 669 | 0.20 | 7-formyl | 469 / 491 | 651 | 0.41 |
| 12-vinyl | 460 | 685 | 0.19 | 12-formyl | 455 | 703 | 0.41 |
| **12-formyl-** | | | | **12-formyl-** | | | |
| 2-vinyl | 442 | 706 | 0.30 | 2-formyl | 454 | 734 | 0.40 |
| 3-vinyl | 427 | 688 | 0.28 | 3-formyl | 455 | 716 | 0.36 |
| 7-vinyl | 442 (sh. 464) | 690 | 0.31 | 7-formyl | 464 | 685 | 0.42 |
| 8-vinyl | 441 | 695 | 0.31 | 8-formyl | 455 | 703 | 0.41 |

The largest spectral changes in the Q-band region were observed for the addition of a second formyl group to 2-formyl- chlorophyll a (chlorophyll f ) and 12-formyl- chlorophyll a. This change causes the Q-band to red-shift by up to 21–69 nm relative to its position in chlorophyll a, and moved the Soret bands by 21–38 nm to lower energies. The lowest energy Q-band (734 nm) was observed at the same time as the largest Soret-Q-band separation (280 nm), for the derivative bearing formyl groups on both the 2- and 12- positions, (12-formyl chlorophyll f ) which can, in principle, be synthesized through the sequential action of the so-called "super-rogue PSII" present in the chlorophyll f biosynthetic pathway (*Ho et al., 2016*; *Trinugroho et al., 2020*) (which introduces a formyl group at position 2 of chlorophyllide a), followed by conversion of 2-formyl chlorophyllide a (chlorophyllide f ) by an engineered synthetic chlorophyllide oxygenase designed to bind chlorophyllide f (instead of its natural chlorophyllide a substrate) in a conformation conducive to methyl

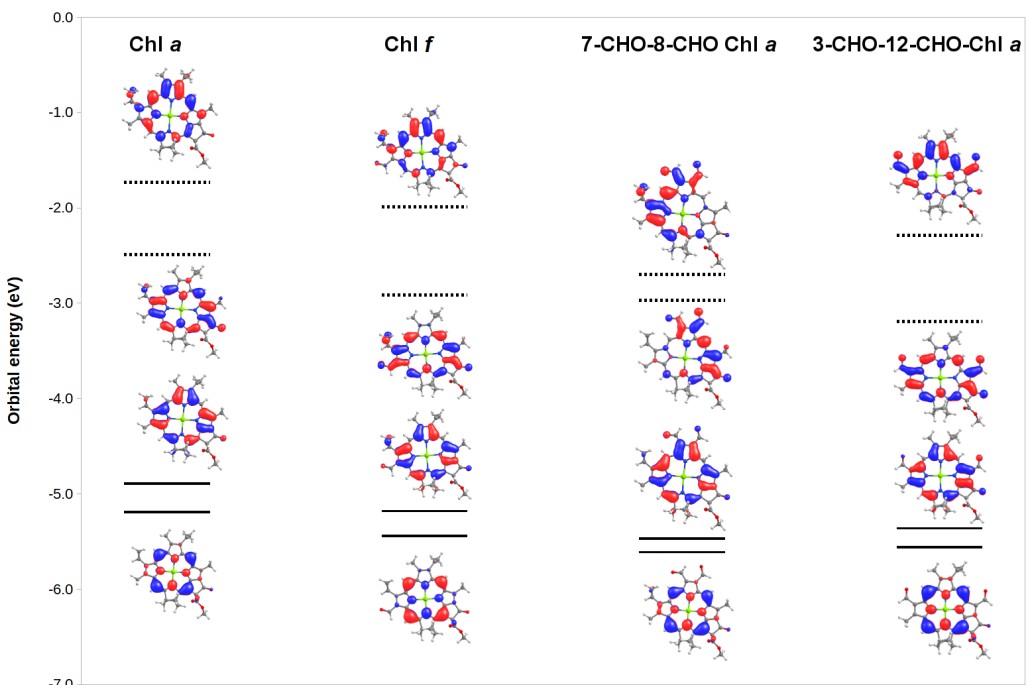

**Figure 5** **MO correlation diagram for representative chlorophyll derivatives.** The calculations employed the B3LYP function, 6-31G* basis set.

oxidation at its $C_{12}$-position. Interestingly, the predicted Soret band for this intermediate is almost as high in wavelength as that of Chl *b*, which suggests that this pigment can have a dual role, both as a constituent (like Chl *b*) of light-harvesting complexes that collect abundant electrons with larger wavelengths than those that can be captured by Chl *a* and a component of photosystem reaction centers attuned to wavelengths even farther into the red portion of the spectrum than those than can be captured by the Soret band of Chl *d*.

To gain additional insight into the origin of the observed spectral changes, we analyzed (Fig. 5) the frontier orbitals of Chl *a* and of three derivatives with extreme spectral changes: 7,8-diformyl-Chl *a* (which has the lowest-energy Soret bands and barely-changed Q-bands), 3,12-diformyl-Chl *a* (which has almost identical red-shifts on both the Q-band and the Soret band), and Chl *f* (which has the one of the two largest Q-band red-shifts among mono-substituted derivatives of Chl *a*). Predictably, both the HOMO and HOMO-1 orbitals are stabilized in the derivatives relative to Chl *a* due to the presence of increased levels of conjugation brought about by the introduction of the additional formyl groups. The red-shifts in Q-bands are shown by this analysis to be due to the fact that in some derivatives the LUMO is even more stabilized by the extra conjugation than the HOMO. Red-shifted Soret bands, in turn, are observed when the ring substituents stabilize the LUMO+1 orbital to a larger degree than the HOMO. The stabilization of the HOMO by formyl groups strongly suggests that oxidation of these derivatives will be more difficult than oxidation of Chl *a*.

Finally, to ensure that the substituted chlorophylls fully retained the ability to function in photosynthetic reaction centers, we computed the redox potentials of their cationic forms. This is a necessary analysis because in photosynthetic reaction centers two electron transfers must be possible: not only must the chlorophyll dimer become a strong reductant upon excitation of its Q-bands, but the cationic form resulting from electron loss from the excited state must retain a sufficiently high redox potential to be able to receive a new electron either from water oxidation by the oxygen-evolving complex (in photosystem II) or from plastocyanin (in photosystem I). The results (Tables 3 and 4) confirm the favorable properties of the substituted chlorophylls, since in all cases, the cationic forms are at least as good oxidizers as Chl *a*, which enables them to oxidize both plastocyanin and the oxygen-evolving complexes. The redox potentials of the derivatives follow much simpler trends than the positions of the absorption bands, and are fundamentally a function of the number of formyl groups present: addition of a single formyl group increases gas phase redox potential by 0.09–0.28 V, whereas presence of two formyl groups increases it by 0.26–0.42 V and changing the number of vinyl groups from one (in Chl *a*) to two or three yields no more than a 0.07 V increase in redox potential.

The increase in redox potential of the $Chl^+/Chl$ pair upon formylation also affects the redox potential of the excited state of the chlorophyl (the $Chl^+/Chl^*$ pair). A simple thermodynamic cycle shows that:

$$E^0(Chl^+/Chl^{excited}) = E^0(Chl^+/Chl) - \frac{excitation\ energy}{F}$$

(where F is the Faraday constant, and excitation energy is the energy of the absorption leading to the excited state), which entails that any increase redox potential of the cation will (if the excitation energy remains the same) also increase the redox potential of the excited state by the same amount and **decrease** the ability of the excited state to act as a reductant. For photosystem I, where the protein environment tunes the reaction center to a potential around 0.5 V (Ishikita et al., 2006), replacement of its Chl *a* pigments by 2-formyl-12-formyl-chloropnhyll would yield a 0.53 V increase in the $Chl^+/Chl$ potential (facilitating electron transfer from plastocyanin) whereas the $Chl^+/Chl^*$ potential would change to 1.03 V − 1.7 V = −0.7 V, which is sufficiently low to enable spontaneous and fast electron-transfer to the physiological electron acceptor (ferredoxin). All other derivatives yield $Chl^+/Chl^*$ potentials below this value, and therefore even more favorable behavior, because they either have a similar (or much smaller) increase in the $Chl^+/Chl$ potential or a more favorable difference in $Chl^+/Chl^*$ and $Chl^+/Chl$ potentials due to the lower wavelengths of their Q-band transitions. For photosystem II, however, the balance of effects is more subtle due to the higher redox potential of their $Chl^+/Chl$ reaction centers (1.25 V) (Grabolle & Dau, 2005) caused by the presence of the oxygen-evolving complex and other local effects imposed by the protein (Ishikita et al., 2006). In this system the thermodynamic cycle above shows that, upon excitation with 680 nm light, the $Chl^+/Chl^*$ potential is approximately −0.57 V, enabling it to quickly and spontaneously transfer its electron to pheophytin (−0.42 V) which ultimately transfers the electron to the PSII quinone (−0.08 V). Replacement of the chlorophylls in the P680 reaction center by

derivatives with higher redox potential and lower excitation energy could therefore easily move the $Chl^+/Chl^*$ redox potential to values above those of pheophytin (which would merely cause a decrease in electron-transfer rate) or even above those of the accepting quinone, which would render the overall electron-transfer in PSII non-spontaneous and therefore much slower (if the derivatives with larger redox potential shifts, like 2,12 diformyl- or 8,12-diformyl-chlorohyll $a$, are used).

The considerations above show that the derivative with the largest red-shift in the Q-band (2,12-diformyl chlorophyll) can be used to increase the number of photons able to power photosystem I for eventual NADPH production or ATP synthesis from cyclic electron flow, but that its use for photosystem II would cause kinetic bottlenecks in electron transfer rates. Derivatives with moderate changes in redox potential are to be preferred for eventual engineering of photosystem II, such as 3-formyl-chlorophyll $a$ (chlorophyll $d$, already found by evolution), 3-formyl-12-vinyl-Chl $a$ ($E^0(Chl^+/Chl^*)$ $=-0.32$ V), or 2,12-divinyl Chl $a$ ($E^0(Chl^+/Chl^*)$ $=-0.49$ V). Replacement of methyl groups (in positions 2-,7-, or 12-) by vinyl groups is expected to be more difficult than the previously proposed formylations at those positions, because the extra carbon atom cannot be readily incorporated by extant enzymes. Instead, it would be necessary to re-engineer the first steps of the protoporphyrin IX biosynthesis pathway so that instead of assembling four equal molecules of porphobilinogen to make hydroxymethylbilane, three molecules of porphobilinogen and one molecule of mutated prophobilinogen bearing two propionate instead of one acetate and one propionate substituent would be used. All subsequent enzymes would then also have to be adapted to accept a 2-carbon substituent at that position instead of a methyl group. It thus appears that improvements on photon capture by photosystem II are currently limited to adapting it to use the naturally occurring chlorophylls $d$ and $f$.

## CONCLUSIONS

The data generated in this study provide a thorough landscape of the spectral and redox changes brought about by vinylation and formylation of the different positions of the chlorophyll scaffold, and how those can be leveraged into eventual use in synthetic biology applications. The most promising possibilities center on the incorporation of 12-formyl chlorophyll $f$ in photosystem I, which dramatically increases the absorption of photons in the low-energy portion of the spectrum ($>650$ nm), and in the large-scale incorporation of Chl $d$ into photosystem II. Several derivatives with longer-wavelength Soret bands with potential applications in engineering light-harvesting complexes with increased absorption range were also found, among which the most promising are 7,8-diformyl-, 2,7-diformyl, and 2,8-diformyl chlorophyll $a$. These three derivatives seem to be reasonably accessible biosynthetically, since incorporation of formyl groups at positions $C_2$- and $C_7$- can be performed by (respectively) the super-rogue PSII or chlorophyllide $a$ oxygenase (*Porra et al., 1994*), whereas formylation at position $C_8$- can be achieved by first preventing (either through inhibition of knocking out the $C_8$-vinyl reductase (*Canniffe et al., 2013*) the reduction of the $C_8$-vinyl group in protoporphyrin IX to ethyl, and then by oxidizing

this vinyl group to formyl (*Loughlin, Willows & Chen, 2015*). We expect that these data will stimulate future endeavors in the development of biological systems with increased photosynthetic efficiency.

### Funding
Work at BIOSIM is supported by the Applied Molecular Biosciences Unit–UCIBIO, which is financed by national funds from FCT (UIDB/04378/2020). The funders had no role in study design, data collection and analysis, decision to publish, or preparation of the manuscript.

### Grant Disclosures
The following grant information was disclosed by the authors:
The Applied Molecular Biosciences Unit–UCIBIO.
National funds from FCT: UIDB/04378/2020.

### Competing Interests
Pedro J Silva is an Academic Editor for PeerJ.

### Author Contributions
- Pedro J Silva conceived and designed the experiments, performed the experiments, analyzed the data, performed the computation work, prepared figures and/or tables, authored or reviewed drafts of the article, and approved the final draft.
- Maria Osswald-Claro performed the experiments, analyzed the data, performed the computation work, prepared figures and/or tables, and approved the final draft.
- Rosário Castro Mendonça performed the experiments, analyzed the data, performed the computation work, prepared figures and/or tables, and approved the final draft.

### Data Availability
All input and output files for the computations are available at Figshare: Silva, Pedro (2022): Redox and spectral properties of substituted chlorophylls. figshare. Dataset. https://doi.org/10.6084/m9.figshare.20747698.v3.

### Supplemental Information
Supplemental information for this article can be found online at http://dx.doi.org/10.7717/peerj-pchem.26#supplemental-information.

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

# PeerJ

acaryochloris marina using chlorophyll d and the effect of chlorophyll exchange. *The Journal of Physical Chemistry B* **126**:4009–4021 DOI 10.1021/acs.jpcb.2c00869.

**Lee C, Yang W, Parr RG. 1988.** Development of the colle-salvetti correlation-energy formula into a functional of the electron density. *Physical Review B* **37(2)**:785–789 DOI 10.1103/PhysRevB.37.785.

**Loughlin PC, Willows RD, Chen M. 2015.** In vitro conversion of vinyl to formyl groups in naturally occurring chlorophylls. *Scientific Reports* **4(1)**:6069 DOI 10.1038/srep06069.

**Milne BF, Toker Y, Rubio A, Nielsen SB. 2015.** Unraveling the intrinsic color of chlorophyll. *Angewandte Chemie International Edition* **54(7)**:2170–2173 DOI 10.1002/anie.201410899.

**Porra RJ, Schafer W, Cmiel E, Katheder I, Scheer H. 1994.** The derivation of the formyl-group oxygen of chlorophyll b in higher plants from molecular oxygen. achievement of high enrichment of the 7-formyl-group oxygen from 18o2 in greening maize leaves. *European Journal of Biochemistry* **219(1–2)**:671–679 DOI 10.1111/j.1432-1033.1994.tb19983.x.

**Schliep M, Crossett B, Willows RD, Chen M. 2010.** 18O labeling of chlorophyll d in acaryochloris marina reveals that chlorophyll a and molecular oxygen are precursors. *Journal of Biological Chemistry* **285(37)**:28450–28456 DOI 10.1074/jbc.M110.146753.

**Schmidt MW, Baldridge KK, Boatz JA, Elbert ST, Gordon MS, Jensen JH, Koseki S, Matsunaga N, Nguyen KA, Su S, Windus TL, Dupuis M, Montgomery Jr JA. 1993.** General atomic and molecular electronic structure system. *Journal of Computational Chemistry* **14(11)**:1347–1363 DOI 10.1002/jcc.540141112.

**Sirohiwal A, Berraud-Pache R, Neese F, Izsák R, Pantazis DA. 2020.** Accurate computation of the absorption spectrum of chlorophyll a with pair natural orbital coupled cluster methods. *The Journal of Physical Chemistry B* **124(40)**:8761–8771 DOI 10.1021/acs.jpcb.0c05761.

**Tanaka R, Tanaka A. 2011.** Chlorophyll cycle regulates the construction and destruction of the light-harvesting complexes. *Biochimica et Biophysica Acta (BBA) - Bioenergetics* **1807(8)**:968–976 DOI 10.1016/j.bbabio.2011.01.002.

**Taniguchi M, Lindsey JS. 2021.** Absorption and fluorescence spectral database of chlorophylls and analogues. *Photochemistry and Photobiology* **97(1)**:136–165 DOI 10.1111/php.13319.

**Trinugroho JP, Bečková M, Shao S, Yu J, Zhao Z, Murray JW, Sobotka R, Komenda J, Nixon PJ. 2020.** Chlorophyll f synthesis by a super-rogue photosystem II complex. *Nature Plants* **6(3)**:238–244 DOI 10.1038/s41477-020-0616-4.

**Voitsekhovskaja OV, Tyutereva EV. 2015.** Chlorophyll b in angiosperms: functions in photosynthesis, signaling and ontogenetic regulation. *Journal of Plant Physiology* **189**:51–64 DOI 10.1016/j.jplph.2015.09.013.

**Weiss C. 1978.** Electronic absorption spectra of chlorophylls. In: Dolphin D, ed. *The Porphyrins (Volume III - Physical Chemistry, Part A)*. New York: Academic Press, 211–223.

**Willows RD. 2003.** Biosynthesis of Chlorophylls from Protoporphyrin IX. *Natural Product Reports* **20(3)**:327–341 DOI 10.1039/b110549n.