# Peer review of "How to tune the absorption spectrum of chlorophylls to enable better use of the available solar spectrum"

_PeerJ Physical Chemistry, doi:10.7717/peerj-pchem.26_

## Round 0.1 · original submission · Major Revisions

Thank you for your submission and your patience with the peer review process. We have now received two quality reviewers of your manuscript. Please address the concerns addressed. Also, please ensure that the abstract and introduction of your paper is accessible and intriguing to non-specialists.

Reviewer 1 ·

Basic reporting

The authors present a computational work on excited state and redox properties of 49 chlorophyll derivatives. They reported vertical excitation energies and redox potentials and proposed their incorporation in the PSI, PSII and other light-activated enzymes. This exciting work presents chlorophyll light-harvesting in a "chemically" logical and coherent way. I have a few points to mention to the authors.

1. Page 2: Authors write, "As a result, the intense absorption bands around 430 nm (the B-bands, or Soret bands) can shift their maxima by up to 15-20 nm, and the Q-bands (around 650 nm) can shift up
49 to 10 nm when the solvent varies from acetone to pyridine."
Can they please specify here the name of the pigment?

2. Page 4, line 112- Why dielectric constant of 20 was chosen? What solvent are the authors trying to model? Also, do authors have performed calculations with a much larger basis set and what is the difference?

3. Table 1: can authors specify the name of the solvent in the caption (experimental)? Also, can the authors justify comparing vertical excitation energies with the absorption maximum? They are not the same things. Absorption maximums are usually "red-shifted" compared to the vertical excitation energies. Can authors also label the excited states? It is unclear if they comparing with the Qy, Qx, Bx or By bands.

4. Line 119: Authors computed the vertical excitation energies, not "absorption bands". The latter one has vibronic features.

5. Line 140: It is not clear to me how a new functional (PBE0) and basis-set (6-31G(d)) appears suddenly when authors mentioned B3LYP/6-31g* in the computational details.

6. Line 262: The redox potential of the radical-cation species in the Photosystem II reaction centre varies as +1.1-1.2 V. Will this affects the authors' proposals? The PSII reaction centre high redox potential is mainly due to the ring orientation (special pair PD1 and PD2), not the OEC or protein effects.

Experimental design

no comment

Validity of the findings

no comment

Reviewer 2 ·

Basic reporting

In this paper, the authors calculated the spectral and redox properties of 49 chlorophyll derivatives, using DFT and TDDFT methods. They reported chlorophyll analogs that absorb light with the red-sifted wavelength. Their results are interesting and provide useful insights into the artificial modification of chlorophyll. However, there is no explanation of mechanism of the wavelength-shift based on the chemistry. I also found the following methodological issues. Because of these reasons, I think additional calculations and analysis are required before acceptance.

Experimental design

(1) DFT Functional
The authors used the B3LYP functional for TDDFT calculations as well as DFT calculations. As the authors recognized, the calculated value of the excitation energy can depend on functional. It was reported that the CAM-B3LYP functional is better than the B3LYP functional in TDDFT calculations of chlorophylls [e.g., M. Higashi et al., J. Phys. Chem. B 118, 10906 (2014). doi:10.1021/jp507259g; K. Saito et al., J. Photochem. Photobiol. A, 358, 422 (2018). doi:10.1016/j.jphotochem.2017.10.003; Zhe Zhu et al., J Chem Phys. 156:124111. doi: 10.1063/5.0083395.]. Therefore, the authors should also calculate using the CAM-B3LYP functional with an appropriate mu parameter.
Because they discussed as “In summary, TDDFT with the PBE0 functional and the 6-31G(d) affords a very good performance in the (low-energy) Q-band absorptions (which are the ones more relevant to the chlorophyl role in photosystems I and II) and a less perfect behavior in the prediction of the features of the (higher energy) Soret-bands”, I guess that they calculated using the PBE0 functional. However, I cannot find the term “PBE0” except for Table 4 in the manuscript. The manuscript seems to be incomplete in describing dependencies on functionals. The authors should mention differences in characteristic of the functionals at least in Introduction and Method.

(2) Fitting equation
(2-1) excitation energy
The authors introduced two independent empirical equations to fit calculated values to measured values:
Soret-band: actual peak (nm)=0.733 × TDDFT-computed absorption peak + 152.8
Q-band: actual peak (nm)= 1.48 × TDDFT-computed absorption peak − 200.0
However, in this equation the fitted value is linier in not the excitation energy but the wavelength which is inversely proportional to the excitation energy. Because previous studies introduced equations that linearly fit the excitation energy rather than the wavelength [e.g., K. Saito et al., J. Photochem. Photobiol. A, 358, 422 (2018). doi:10.1016/j.jphotochem.2017.10.003; K. Saito et al., J. Photochem. Photobiol. A, 402, 112799 (2020), doi:10.1016/j.jphotochem.2020.112799], the authors should check availability of the fitting in the excitation energy and compare it with the fitting in the wavelength.
(2-2) redox potential
Similar to the analysis of the excitation energy, a fitting equation may be needed to fit the calculated absolute value to the measured value [e.g., Kishi et al. , Photosynth. Res. 134:193-200 (2017). doi: 10.1007/s11120-017-0433-4] At least, the authors should provide a plot of the measured versus the calculated values of redox potential like Figure 1 for the excitation energy, and propose the fitting equation for redox potential If necessary.

(3) Solvation effect
As “Redox potentials were computed from the difference in energies between the geometry-optimized neutral molecules and their (geometry-optimized) one-electron-oxidized forms at the B3LYP/6-31G(d) level using the polarizable continuum method implemented in Firefly, with a dielectric constant of 20”, the authors took into account the solvation effect using the PCM in the calculation of redox potentials. How did they treat the solvation effect in excitation energy calculations? In the comparison with the measured spectra (Tabel 1 and Figure 3), the authors should mention the solvent used in the experiment and compare the calculated value using the corresponding PCM model.

Validity of the findings

Mechanism of wavelength shift.
“Introduction of a vinyl group at positions 2- or 3- shifts both the Soret band and the Q-band to lower energies (higher wavelengths).”
“At positions 7- and 8-, both vinylation and formylation blue-shift the Soret band”
“In all studied positions, vinylation increases the absorption wavelenght of the Soret-band by moderate amounts (6-13 nm) but only causes significant shifts (13-17 nm) at the Q-band absorption when it occurs at the 2- or 12- position. In contrast, and depending on the modification site, formylation has widely disparate effects on both the Soret and Q-bands”
What is the chemical mechanism of the wavelength shift? I believe that the mechanism should be explained by the change in the pi-conjugated system of chlorophylls. Because the Soret and Q-bands can be assigned qualitatively to transitions between the four orbitals (HOMO, HOMO-1, LUMO, and LUMO+1), i.e. the so-called four orbital model, I suggest the authors to show the energy levels and the shapes of HOMO and LUMO (also HOMO-1 and LUMO+1 if necessary). I expect, for example, that the origin of the redshift in the wavelength is a decrease in the LUMO energy or an increase of HOMO energy owing to the change in shape of the molecular orbital on pi-conjugated system. The authors should try to explain the chemical mechanism of the wavelength shift.

---

## Round 0.2 · Minor Revisions

The manuscript is much improved. Please further clarify the points raised by Reviewer 2. Thank you.

Reviewer 1 ·

Basic reporting

The authors have now made the necessary changes in the manuscript. I am happy to recommend this work for publication.

Experimental design

no comment

Validity of the findings

no comment

Reviewer 2 ·

Basic reporting

The authors have revised the manuscript appropriately. I think that the following results with figures related to the TDDFT calculations with PCM model should be provided in Supporting Information. I am afraid that many readers may wonder why the authors did not consider the solvent despite their comparison with observations in organic solvents. Thus, it is an important conclusion of this study that the calculation in vacuum is sufficient in this case.

“We have, however, now performed those computations for chlorophylls a, b, d, and f. The predicted spectra have more structure than the ones computed in the gas-phase, but the fit of absorption maxima has worse R2-values than the one obtained with the gas-phase results (see figures below). Therefore we think that the strategy suggested by the reviewer, while a priori sound, end up not affording a sufficient improvement above our chosen methodology.”

Experimental design

no comment

Validity of the findings

no comment

---

## Round 0.3 · accepted · Accept

Thank you for your patience and cooperation with the review process. I believe the final version is a stronger manuscript, and will draw more readers.